# Integrating quantitative and qualitative methodologies to build a national R&D plan using data envelopment analysis based on R&D stakeholders' perspectives

**Wan Park[1], Sang-Gook Kim [2]***

**1** Science and Technology Management Policy, University of Science & Technology, Daejeon Metropolitan City, Republic of Korea, **2** R&D Investment and Strategy Research Center, Korea Institute of Science and Technology Information, Seoul, Republic of Korea

\* sgkim@kisti.re.kr

**Data Availability Statement:** All relevant data are within the manuscript and its Supporting information files.

## Abstract

The yearly increase in government R&D investment and top-down national R&D investment allocation requires a more quantitative decision-making system that maximizes R&D performance and efficient budget allocation. Sound decision-making is necessary at both the selection stage and the pursuit stage in order to maximize limited national R&D resources. We study Korean smart farms as an example to examine national R&D investment from the various R&D actors (academia, industry, and research institutes) perspectives. The objective of our research is to evaluate the theoretical efficiency of R&D investment on specific technologies in smart farms and compare the results with expert opinions where the reality is reflected. To be specific, our study is to provide the quantitative approach in making decision among policymakers by reflecting the field experiences and opinions. We use a data envelopment analysis with an assurance region model, which integrates an analytic hierarchy process and a data envelopment analysis. The weights of output in DEA model by the R&D actors are similar to the overall weight by all actors, implying that investment allocation decisions in the smart farm sector are not significantly affected by the R&D actors. We realized that the relative efficiency of some R&D technologies increases after reflecting qualitative ideas of experts. In reality, it is necessary to invest in these technology groups, but they excluded from top-down decision-making. This also shows that a government's top-down decision-making can distort its investment allocation. This study proposes a new approach to compensate for the difference between theoretical virtual prices and actual prices in data envelopment analysis. In particular, when comparing the only quantitative results on investment priorities with analysis results by additionally reflecting the opinions of experts in each sector, we found that the Korean government's investment priorities in the smart farm field are considerably distorted. Therefore, this study is expected to be used as an alternative for policy makers to compensate for the quantitative distortion might be caused by top-down national R&D investment decisions.

**Funding:** The authors received no specific funding for this work.

**Competing interests:** The authors have declared that no competing interests exist.

## Introduction

Professor Paul M. Romer, a laureate of the 2018 Nobel Prize in Economic Sciences, argues that technological progress does not fit the classical growth model, which is determined by factors external to the model, that is, actions (e.g., Research and Development (R&D)) encouraged through economic incentives. This is referred to as the endogenous growth model [1]. According to Romer [2], unlike public and other existing goods, technology (or knowledge) is non-rival, which means it can be shared by many people without depleting its availability. Nevertheless, it is also excludable, and can be protected by laws or institutions to limit its use. The higher the excludability, the greater the private return from R&D, which further encourages such activities in the private sector (or industry). However, knowledge that is less likely to be excluded, such as basic science, is usually examined in universities or public research institutions. These characteristics allow for appropriate government intervention in order to achieve an efficient resource allocation. Therefore, the main purpose of national investment in R&D is to compensate for private R&D activities, support the development of future core technologies, and develop technologies in the basic, public, and welfare sectors [3]. As the national investment resources are limited, it is critical to make a priority which research needs support or technologies are developed. It requires government to challenge how to utilize the available but limited resources best for its science and technology research. In this regards, the priority setting is important when it comes to national R&D investment. According to OECD [4], OECD member governments are tackling decision making in priority setting for R&D. The report defines priority setting as a complex political process involving many people from various fields. It includes from thematic to structural priorities such as training researchers or balancing the funding options. The decision-making for setting the R&D investment priority is very complicated process which involves various stakeholders from scientific experts to policy, business and community representatives.

Korea's investment in R&D has increased significantly, with the country ranked fourth largest in the world in 2021, and top in terms of percentage of GDP. Although investment in national R&D is increasing, research performance remains poor, given the size of the budget. In addition, the innovation performance in R&D system in Korea is still below the average among OECD countries. The reason is that the R&D resource allocation is mainly under control of government and the decisions are made by top-down process (e.g., government designs the project and the companies or institutions are applied the project) qualitatively. However, it is difficult to change the current decision-making process immediately. Therefore, this increase, and the top-down national R&D planning it needs, requires a more quantitative R&D investment decision-making system that can maximize technology transfer and diffusion and yield an efficient budget allocation. Given the country's limited resources, national R&D projects should undergo a selection process to maximize their effectiveness.

Policymakers responsible for allocating the national R&D budget need to make optimal investment decisions in order to maximize research performance. Several studies have examined the performance and efficiency of R&D investment, and proposed methodologies to solve these problems. Jang [5] develops and applies a national R&D budget allocation model based on machine learning and optimization theory. Lee et al. [6] use a data envelopment analysis (DEA) to analyze the relative efficiency of investment in the field of precision medicine. In addition, Kim et al. [7] analyze the effect on investment performance of R&D actors, development levels, and periods, using technology transfer and diffusion as their performance measure for national R&D investment. Wang and Huang [8] evaluate the relative efficiency of

R&D activities in 30 countries using DEA method. They show that 15 countries are fully efficient in R&D activities and more than 20 countries are increasing their returns to scale. They also find out that there is chance to improve the effectiveness of policies and efficiency in R&D performance. Kyung [9] proposes a national R&D investment priority model based on a technology roadmap. At the national level, the roadmap is used to secure the competitiveness of domestic technology, and to determine the directions and priorities for resource allocations. In the process of identifying technologies in which to invest, the development of convergent technologies can be reflected in the opinions of R&D actors. As a result, governments can effectively allocate and use R&D resources by selecting and focusing on certain technologies. In recent years, the DEA method has become popular for analyzing the efficiency and performance of R&D investments, at both the national and the industry level. Liu et al. [10] evaluate the R&D performance of industries in China with two-stage DEA model. Contreras and Lozano [11] solve the problem of additional resources allocation among public universities by utilizing centralized DEA with bargaining theory. Khoshnevis and Teirlinck [12] evaluate the performance of R&D active firms to allocate R&D resources efficiently. They utilize the DEA model with ratio inputs and outputs. Ozbugday et al. [13] analyze OECD countries to distinguish the efficiency changes in long-term. They implement input oriented DEA with Malmquist Productivity Index. Most of previous literatures utilize DEA as a tool to analyze the performance of various sectors including R&D performance. They focus on the R&D performance according to DEA results.

In this study, we develop a scale that can measure the efficiency in producing various outcomes compared to the R&D expenditure input and the relative importance (or preference) qualitatively considered for individual performance indicators by industry, academia, and research institutes. Therefore, we present a new model in the form of a mixture of qualitative and quantitative analysis that can measure the efficiency of input versus performance within the importance range for the intended output variables by preemptively measuring them. At this time, evaluating in advance the importance (or preference) of performance indicators by each innovation entities (i.e. industry, academia, and research institutes) in the smart farm sector in Korea through AHP and reflecting the additional constraints on the coefficients of the output variables in the DEA model with an assurance. Lastly, in order to secure the reliability of the optimal investment alternatives based on the newly proposed DEA-AR model, sensitivity analysis is performed to analyze the change in the appropriate amount of input that will not change the optimal investment priority. As a result, we evaluate whether national R&D investment decisions reflect reality rationally and objectively. And we use the results to derive policy implications. This study in this regard aims to provide better quantitative decision making for policymakers when they allocate resources on national R&D.

The remainder of this paper is organized as follows. Section 2 outlines the analytical framework for R&D investment priorities with the research method. Section 3 gives a description of the data and the mathematical modelling that we use in the paper. Section 4 presents the empirical results and analysis of Korea's smart farm R&D investment priorities. We apply the proposed model to compare the current investment priorities with investment areas considered important to performance, as identified by relevant R&D actors. The final two section provides the discussion and conclusion of the results which show the empirical, practical implications and academic contributions for future policymakers.

## The analytical framework for R&D investment priorities

**An overall summary of the methodology.** To quantitatively evaluate investment decision-making related to national R&D planning, we apply an integrated an analytic hierarchy

process (AHP) and DEA model to analyze a DEA-AR, with weights constrained using the results of the AHP. The DEA is a nonparametric decision-making methodology that objectively determines relative efficiency based on minimal information. One of the biggest advantages of the DEA is that the model generates the values or weights of the inputs and outputs automatically, rather than relying on analysts to provide such information in advance [14]. However, most observations have an efficiency value of one, making it impossible to compare the efficiency of the evaluation or to limit the number of projects determined to be the most efficient [15]. Therefore, once an absolute value judgment is available from analysts, the results can be enhanced using real data by reflecting the value judgment in advance. Various methods have been proposed to improve the discriminating power of the DEA model. One such method restricts the range of weights used to indicate the relative importance of the values in the DEA model. Thompson et al. [16] recognize the weaknesses of the DEA model, explicitly applying an AR in their efficiency analysis of the 100 largest US banks. R.G.Dyson and E.Thanassoulis [17] formulate a weight-restricted AR model, and suggest various types of weight restrictions and value judgments for the AR. Allen et al. [18] summarize the development of weight restrictions and provide a basic overview of the model. They also discuss in detail direct weight constraints, adjusting the input and output variables, and actual input and output variable constraints.

In order to calculate the weight for each R&D actor in terms of R&D performance, we categorized the actors as belonging to industry, academia, or a research institute, as in previous studies. Each R&D actor's AHP value was then used as a weight constraint in the performance indicators. The conceptual structure of the analytical model is shown in Fig 1.

**Measuring key indicators to evaluate the performance of government decision-making.** The AHP is a decision-making method that uses pairwise comparisons to select or prioritize one of several alternatives. The computational model was originally devised by Prof.

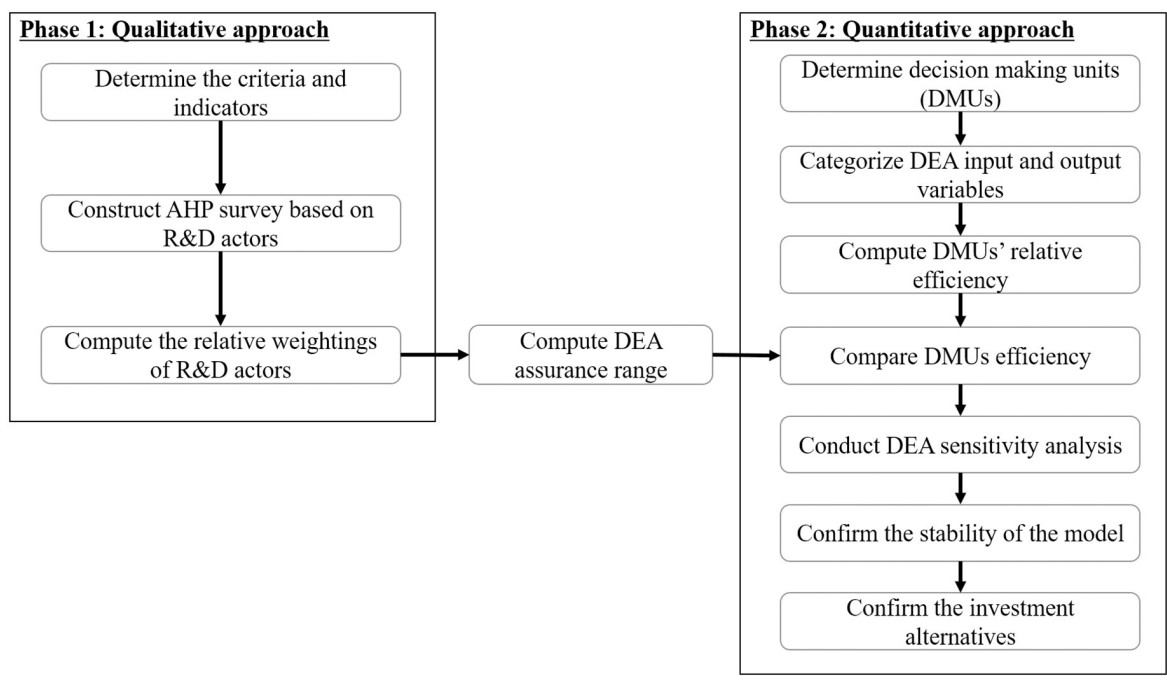

**Fig 1. The conceptual analytical model.**

Thomas L. Satty to analyze problems hierarchically, similarly to the way people do. The model is widely used in the field of decision-making because it provides a quantitative measure (a ratio) of relative importance. Here, we employ an AHP to obtain key indicators, which we use to evaluate the performance of government decision-making related to prioritizing R&D investment in specific sectors.

**DEA model with restricting the weight ranges.**   Efficiency can be measured in several ways. The DEA model measures the efficiency of output over input nonparametrically. Based on linear programming, the empirical efficiency frontier is derived from the outputs and inputs, and then each Decision Making Unit (DMU) measures its inefficiency as its distance from the efficient frontier [19]. The DEA model was developed in 1978 by Charnes et al. [20] combining Farrell's efficiency analysis theory from 1957 and Shephard's distance function theory from 1970. Various extensions to the DEA model have been proposed, including the Charnes–Cooper–Rhodes (CCR) model and the Banker–Charnes–Cooper (BCC) model [21]. The CCR model assumes constant returns to scale. The BCC model, which extends the CCR model, assumes variable returns to scale, which change with the scale of inputs and outputs. In addition, the DEA model is divided into input- and output-oriented models. The input-oriented model derives an improvement plan that minimizes inputs and maximizes output efficiency. The output-oriented model maximizes output for optimal input efficiency [14].

The DEA model determines the relative efficiency of various areas, based on minimal information. However, the conventional DEA model has practical limitations: it lacks discriminating power, and it cannot limit the number of efficient projects. This may be problematic in practice, where the number of government R&D projects is usually limited by budgetary constraints. If the number of efficient projects identified by the DEA analysis is larger than the budget, there is no way of determining which projects are superior [15]. To solve this problem, methods have been proposed that restrict the range of a weight. In contrast to the conventional DEA model, the DEA-AR model, developed by. Thompson et al. [22] performs an additional analysis on the DMUs on the efficiency frontier. The difference between this and the DEA model is the weight constraint formula. The weight range sets upper and lower limits for the ratio between the input and output weights.

Fig 2 is a two-dimensional illustration of the DEA-AR model. It considers one input variable ($X$) and two output variables ($Y$) and assumes that each DMU is given by the points in the figure. Then, the efficient DMUs are P2, P3, P4, and P5. Because there is only one input variable, only the weight range of the output variables needs to be considered. This weight range is measured from the origin, and these lines become the constraint range, or the AR. When constraining the AR range from R1 to R3, DMUs P3 and P4 (within the range of R1 to R3) become efficient. Similarly, if we constrain the AR range from R1 to R2, P3 becomes the only efficient DMU. As described above, by constraining the AR, we obtain the most efficient DMU, thus solving the problem of a lack of discriminating power. Several methods exist for setting the AR range by incorporating expert opinions directly, using a conventional decision-making analysis or a multiple regression analysis. In this study, we set the weight range using expert opinions obtained from an AHP analysis.

**Integrated models with assurance region.**   The integrated AHP/DEA model overcomes the weaknesses of the individual models, constraining the AR range using the AHP. Ho [23] reviews four literature articles regarding integrated AHP/DEA approach and discovers that AHP/DEA model has less studied comparing with other AHP combined model, such as mathematical programming techniques (i.e. linear programming). Several studies have introduced the empirical applications of the integrated AHP/DEA methodology. Wang et al. [24] apply the integrated model to assess the bridge risk in the UK. AHP is applied to calculate the weights of four criteria (safety, functionality, sustainability and environment) with linguistic

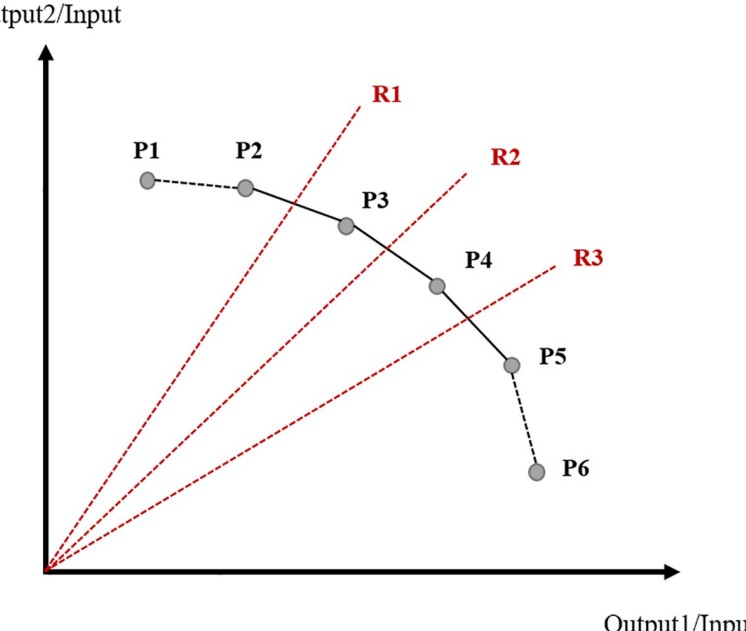

**Fig 2. A schematic representation of the DEA-AR model.**

terms (high, medium, low and none) by top manager of bridge maintenance projects and then DEA utilizes to determine the values of the linguistic terms. Yang and Kuo [25] propose AHP for evaluating qualitative DMUs and DEA for calculating both qualitative and quantitative data in order to design the efficient plant layout. Sueyoshi et al. [26] combine AHP and DEA to help making efficient decision for internal auditing. AHP is used to deal with qualitative information from professionals and DEA is applied to compute the quantitative data from the accounting department and the fleet information. Lin et al. [27] propose an integrating DEA and AHP model to evaluate the economic performance in China. Among ten variables, two qualitative variables are measured by AHP and used in DEA analysis while other quantitative variables are used in the DEA analysis. Tapia et al. [28] develop a hybrid AHP and DEA framework to select efficient carbon dioxide utilization options. They apply the model to two case studies by combining quantitative data and expert opinions on qualitative criteria as a priority weights. Jang and Lee [19] verify the practical application of an analytical hybrid model to select an optimal promising technology, given limited national R&D resources. Kim and Choi [29, 30] and Park [31] evaluate the efficiency of container terminals using a hybrid AHP/DEA model. Lee et al. [32] use an integrated AHP/DEA model to develop a realistic mathematical model that can be applied to select R&D projects. Sinuany-Stern et al. [33] complement the limitations of each decision model by ranking DMUs using an AHP, combining the AHP models with a DEA. In addition, Rhim et al. [15] propose a method for determining the AHP model in a hierarchical structure, and then using the values as weight constraints in a DEA model. They present a case study that uses an integrated AHP/DEA model to evaluate and select R&D projects. Ramanathan [34] derives a comparative matrix using a DEA to design a method for ranking alternatives using an AHP. Wang et al. [35] set weight ranges for the inputs and outputs and suggest an alternative model that compensates for the limitations of the AHP/DEA model. Lee et al. [36] compares the performance of national R&D programs using DEA and DEA-AR model to incorporate the importance of variables in reality. The

pairwise comparison in the AHP for the upper and lower bound of AR was introduced to measure the priority weights of output variables. Lee and Lee [37]also introduce AHP to make the pairwise comparison for DEA-AR model, setting weights of upper and lower bounds. Lai et al. [38] use AHP/DEA-AR model for evaluating the efficiency of international airports. AHP results are used to calculating the upper and lower bound of DEA-AR. There are various application studies with integrated model from business sector to transportation. Most of studies use AHP as evaluating qualitative DMUs or ranking the variables of input and output.

**Sensitivity analysis for confirming optimal alternatives.**   The calculation cost issue in the DEA model is highly related to the number of input and output variables. In the investment efficiency analysis model for national R&D projects proposed in this study, the number of input and output variables is generally not large. Similarly in the case of Korean smart farm, the input variable is R&D funds, and the output variable is composed of seven performance indicators representing technological and economic performance. Due to the nature of the DEA model, there may be a stability issue regarding the optimal efficiency score according to the amount of change in the input variable that tried in this research. There is a need to confirm the maintainability of the optimal solution of the DEA-AR proposed model within a specific change range of the input. What this means is that it is necessary to check whether the complementary slackness theory in the DEA liner programing is satisfied according to the change in the input amount. Within the range of the amount of change of the input that can satisfy the complementary slackness condition, the conversion between the primal and the dual in the linear programing becomes easy, and there is a possibility that the calculation cost of the linear programming method will be lowered. Therefore, in this study, sensitivity analysis on the DEA model is additionally performed to solve these two issues.

## Data and mathematical modelling

**DMU.**   The DMUs in this study are 26 smart farm technology groups, which we selected by grouping similar technologies using document clustering, based on SCOPUS data. Then, the final technologies were selected by collating the expert opinions. The smart farm DMUs are given in Table 1.

**AHP modelling.**   For the AHP survey, we selected two categories, containing seven indicators in total, given the availability of data and their ability to calculate the weights for the experts. Government investment in R&D is important to the effectiveness of core technologies in industry, and causes economic ripple effects in investment; therefore, we categorize the criteria as technical and economic effects. Technical effects include papers, patents, and technical link indicators; economic effects include the employment inducement effect, value-added inducement effect, forward and backward industry link effects from the Bank of Korea, and input-output statistics. Technical and economic effects are located directly below the primary hierarchy target, that is, "priorities of national R&D investment," followed by a further seven sub-criteria. The AHP model and criteria for this study are given in Table 2.

**DEA input and output variables.**   The inputs and outputs are based on AHP criteria derived in previous works on R&D investment efficiency. The model is implemented with one input variable (the cumulative amount of R&D investment from 2013 to 2018) and seven output variables (papers, patents, employment inducement effects, value-added inducement effects, and forward and backward industry effects). The inputs and outputs identified by previous studies on R&D investment efficiency are shown in Table 3.

**Mathematical linear programming: DEA and proposed DEA-AR model.**   This study analyzes the input-oriented CCR model using the benchmarking package in R. The assumption of DEA model is that there are $i = 1,...,m$ inputs and $r = 1,...,s$ outputs. The overall

**Table 1. Smart farm technology group (DMUs).**

| DMU | Technology Group |
|---|---|
| T1 | Environmental Condition Monitoring |
| T2 | Smart Complex Environmental Controls |
| T3 | Watering System Measurement |
| T4 | Smart Irrigation System |
| T5 | Cooling/Heating AC System Control |
| T6 | Biometric Information Measurement |
| T7 | Smart Machinery |
| T8 | Pest and Disease Diagnosis |
| T9 | Pest and Disease Management |
| T10 | Smart Product Sorting |
| T11 | Long-term Storage Packaging Technology |
| T12 | Vertical Farm Building Design |
| T13 | Multi-sensing AI Control |
| T14 | Artificial Synthesis |
| T15 | AI Farmbot |
| T16 | Hybrid Renewable Energy |
| T17 | Organic Water Reclamation |
| T18 | Customized Functional Ingredient Control |
| T19 | Nano-bio-sensing |
| T20 | Integrated Environmental Information Processing Platform |
| T21 | Crop Growth Big Data Platform |
| T22 | Productivity Optimizing Management Platform |
| T23 | Smart Quality Control and Distribution Platform |
| T24 | Printed Farm |
| T25 | Advanced Production |
| T26 | Smart Food |

**Table 2. Criteria for AHP related to national R&D investment priorities.**

| Level 1 | Level 2 | Definition |
|---|---|---|
| Technical Effect | Paper | Trend frequency in similar papers published by technology groups in the past five years (2014–2018) |
| | Patent | Trend frequency in patent applications and registrations of similar prior arts by technology groups in the past five years (2014–2018) |
| | Technical Linkage | Strength of a linkage between candidate technology groups |
| Economic Effect | Employment Inducement Effect | The amount of labor incurred directly or indirectly in the process of production, distributed according to the demand in the specific industry (includes job-inducing effects) |
| | Value-added Inducement Effect | The added value (operating profit after tax less capital costs) resulting from production in related industries, along with an increase in the demand for units of the specific industry |
| | Forward Industry Link Effect | The effect on the output of a specific industry (or all industries), based on the change in demand for that industry (or all industries) |
| | Backward Industry Link Effect | |

**Table 3. Inputs and outputs identified in existing studies.**

| Input | Output | References |
|---|---|---|
| • Forward industry link effect | • Employment creation effect | Lee et al. [6] |
| • Backward industry link effect | • Value-added creation effect | |
| • Government R&D investment | • Number of Korean patents | |
| | • Number of Korean papers | |
| • Investment | • Discounter cash flow | Eilat et al. [39] |
| | • Customer focus group feedback | |
| | • Performance improvement metrics | |
| | • Congruence | |
| | • Importance | |
| | • Synergy with other operations | |
| | • Propriety position | |
| | • Platform for growth | |
| | • Durability | |
| | • Probability of tech & commercial success | |
| • 1st year investment | • Papers | Chun et al. [40] |
| • 2nd year investment | • Patent registration | |
| • 3rd year investment | • Patent application | |
| | • Attending | |
| | • Conference | |
| • Gross expenditure on R&D | • Patents | Wang and Huang [8] |
| • R&D capital stocks | • SCI/EI papers | |
| • Researchers | | |
| • Technicians | | |
| • R&D capacity | • Technology expandability | Jang and Lee [19] |
| • Producibility | • Marketability | |
| • Investment | • Expected profit | Lee et al. [32] |
| • Duration | • Technological spillover | |
| • R&D investment | • Number of patents | Kim et al. [41] |
| • Government subsidies | • Supply level | |
| | • Production cost | |
| • Funds | • Domestic non-SCI papers | Lee et al. [36] |
| • Researchers | • International SCI papers | |
| | • Domestic applied patents | |
| | • Domestic granted patents | |
| | • Foreign applied patents | |
| | • Foreign granted patents | |
| | • Master's degree students | |
| | • Doctoral degree students | |
| • R&D expenditure | • Number of patents | Han et al. [42] |
| | • Number of papers | |
| • R&D expenditure | • Patent applications | Zhong et al. [43] |
| • Full-time equivalent of R&D personnel | • The sales revenue of new products | |
| | • The profit of primary business | |

efficiency value of a particular DMU can be evaluated using following equation where it should be noted that the variables are the $u_r$'s and the $v_i$'s and the $y_{rk}$'s and $x_{ik}$'s are the observed output and input values, respectively, of $DMU_k$.

$$maxZ_k = \sum_{r=1}^{s} u_r y_{rk}$$

$$\text{s.t. } \sum_{r=1}^{s} u_r y_{rj} - \sum_{i=1}^{m} v_i x_{ij} \leq 0 (j = 1, \ldots, n) \tag{1}$$

$$\sum_{i=1}^{m} v_i x_{ik} = 1$$

$$u_r \geq 0 (r = 1, \ldots, s) \, v_i \geq 0 (i = 1, \ldots, m)$$

The DEA-AR model of this study is expressed as follows:

$$maxZ_k = \sum_{r=1}^{7} u_r y_{rk}$$

$$\text{s.t. } \sum_{r=1}^{7} u_r y_{rj} - vx_j \leq 0 (j = 1, \ldots, 26) \tag{2}$$

$$vx_k = 1$$

$$u_r \geq 0 (r = 1, \ldots, 7), v \geq 0$$

$$L_i u_1 \leq u_i \leq U_i u_1 (i = 2, \ldots, 7) \tag{3}$$

where $Z_k$ represents efficiency of $DMU_k$ to be evaluated and $u_r$ is the weight of output $r$ (*paper, patent, . . ., backward industry link effect*) while $v$ is the weight of input (*cumulative R&D budget*). For the value of output and input, we denote $y_{rj}$ is the value of output $r$ (*paper, patent, . . ., backward industry link effect*) for $DMU_j$ and $x_j$ is the value of input (*cumulative R&D budget*) for $DMU_j$. The DEA-AR model put restrictions on the upper bound ($U_i$) and lower bound ($L_i$). Lower bound for the ratio of virtual price in output variable is $u_i/u_1$ while upper bound for the ratio of virtual price in output variable is $u_i/u_1$.

In this study, we apply AHP to set the relative weights for DEA assurance range by mirroring real world opinions. First, experts from each fields (industries, academia, and research institutes) independently make their own pairwise comparison matrices and derive the relative weights for output variables. Then, the upper ($U_{1i}$) and lower ($L_{1i}$) bounds for assurance region are determined by the relative weight ratio of output variables as

$$L_{1i} = min\frac{u_i}{u_1}, U_{1i} = max\frac{u_i}{u_1} \quad (i = 2, \ldots, 7) \tag{4}$$

where $u_1$ is the weight vectors of the 1^st output indicating *paper* while $u_i$ is the weight vectors of the $i$^th output which is the one of the remaining output variables but *paper*. When the absolute values of the upper bound ($U_{1i}$) and the lower bound ($L_{1i}$) analyzed through AHP analysis are greater than or equal to 1 and less than 1, the consistency of relative weight is broken. In this case, when adding the AR constraints, there may be a case where it is necessary to guarantee the consistency between the upper bound and lower bound, and a case where it is not. AR

constraints were set without correcting consistency in this study to reflect all opinions of individual experts in the smart farm sector.

**Sensitivity analysis.** Variation of the input within the range that maintains an optimal solution in DEA (namely, the radius of stability) does not change the efficiency of the DMU. If the complementary slackness condition is satisfied, the newly proposed DEA-AR model can have a dual linear problem, and the optimal solution in this case is the same. Therefore, if the complementary slackness condition is satisfied, there a chance for the computation cost to be reduced as much as the number of variables of the objective function in the dual problem. In this study, sensitivity analysis was performed to check whether the complementarity slackness condition was satisfied. According to a previous study [44], it was pointed out that the sensitivity analysis method used in the existing linear programming method is not suitable for use in DEA. It suggests the need to perform sensitivity analysis while checking whether the efficiency of each DMU belonging to the efficiency group and the inefficiency group changes according to the amount of change in the input variable. Therefore, in this study, by referring to previous studies, a sensitivity analysis was performed to show how stable the analysis results are while changing the input values little by little. That is, the efficiency and inefficiency DMU were classified and the stability of the analysis result was confirmed by changing it from 5% to 20% based on the optimal input amount. As a result, even in the smart farm case of this study, the complementary slackness condition was satisfied within the range of 5% to 20% of the optimal input amount, confirming the possibility of retaining the dual problem. In this case, the number of variables in the given constraint and DEA-AR problem is not large, and the additional constraint constrained by the assurance region only limits the feasible range, so the calculation cost is not high even for the primal problem, not the dual problem.

## Empirical results of the R&D investment priorities by R&D actors

**Weight range of key performance indicators considered significant in R&D planning and strategies by R&D actors (total, industry, academia, research institutes).** Table 4 shows the weights of the indicators viewed as important to R&D planning by experts representing industry, academia, and research institutes. In general, we can suppose that academia is more interested in the fundamental or basic research, whereas industry is more interested in development research; research institutes bridge the two. Given the importance of the key indicators identified by actors in terms of their technical effect, papers and patents are considered relatively essential by industry, and technical links are considered important by academia. In terms of the economic effect, research institutes consider the employment inducement and value-added inducement effects to be substantial, whereas academia values the forward and backward industry link effects. These results indicate that academia and research institutes

**Table 4. Comparison of weights between R&D actors.**

| Level 1 | Level 2 | Total | Industries | Academia | Research Institutes |
|---|---|---|---|---|---|
| Technical Effect | Paper | 7.00% | 8.17% | 5.61% | 6.99% |
| | Patent | 8.41% | 9.82% | 6.55% | 8.56% |
| | Technical Link | 12.57% | 12.69% | 13.25% | 11.88% |
| Economic Effect | Employment Inducement Effect | 13.28% | 12.15% | 12.65% | 14.93% |
| | Value-added Inducement Effect | 27.66% | 26.54% | 24.15% | 31.71% |
| | Forward Industry Link Effect | 15.55% | 15.33% | 18.92% | 12.96% |
| | Backward Industry Link Effect | 15.55% | 15.33% | 18.92% | 12.96% |

**Table 5. AR upper and lower range from the AHP analysis.**

| Actors | Total | | Industries | | Academia | | Research Institutes | |
|---|---|---|---|---|---|---|---|---|
| AR range | Upper | Lower | Upper | Lower | Upper | Lower | Upper | Lower |
| $u_2/u_1$ | 2.2 | 0.6 | 2.0 | 1.0 | 2.2 | 0.6 | 2.1 | 1.0 |
| $u_3/u_1$ | 7.9 | 0.8 | 6.4 | 1.0 | 7.9 | 0.8 | 7.1 | 1.0 |
| $u_4/u_1$ | 7.5 | 1.0 | 2.3 | 1.0 | 6.5 | 1.2 | 7.5 | 1.0 |
| $u_5/u_1$ | 33.6 | 1.0 | 8.3 | 1.0 | 33.6 | 2.0 | 19.3 | 1.0 |
| $u_6/u_1$ | 13.4 | 0.5 | 12.4 | 0.5 | 12.3 | 0.5 | 13.4 | 0.5 |
| $u_7/u_1$ | 13.4 | 0.5 | 12.4 | 0.5 | 12.3 | 0.5 | 13.4 | 0.5 |

consider the weights of application and development to be higher than those of basic research, because smart farms tend to implement convergence technologies.

The range of the DEA-AR weight constraint is as follows (Table 5). The range for each R&D actor is set to 0.5 to 33.6, which was derived in a similar way to the overall weight, including all R&D actors. Once again, the AHP weight results by R&D actor reflect characteristics of the smart farm industry, as shown in Table 4. Therefore, we conclude that because most smart farm technologies are convergent technologies, they do not vary between individual R&D actors, as expected.

**Comparative analysis of unconstrained DEA and weighted DEA-AR.** Prior to the comparative analysis, organic water reclamation (T17) and printed farm (T24) had no input values; therefore, we exclude them from the DEA and DEA-AR analysis. Fig 3 indicates the results of efficiency scores in the conventional DEA model without weight constraints, and the DEA-AR model with weight constraints, applying each R&D actor's weight and the overall weight. Once the AR constraints are added, the efficiencies of DMUs can be lower than that of the unconstrained DEA. In this study, most DMUs, reflected in the weight of each R&D actor, were less efficient than before. This is because as the weight constraints are added, the efficiency area narrows, becoming smaller than the maximum value of the objective function of the general CCR model. The DEA-AR results indicate that there is no significant difference in the weight

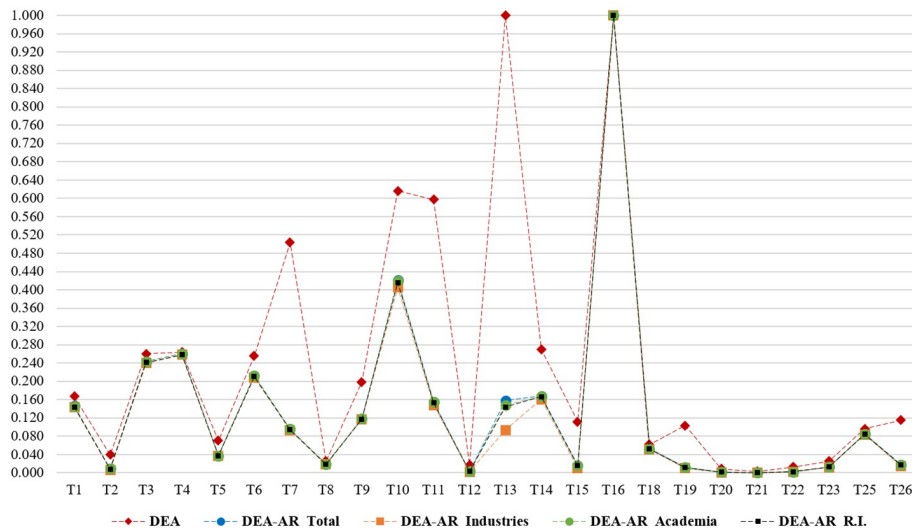

**Fig 3. Comparison of efficiency scores between the DEA and DEA-AR models.** Note: DMUs T17 and T24 are excluded because their input values are zero. R.I represents Research Institutes.

range for each R&D actor. This means that they have similar views in practice given the technologies of smart farm that applied. As noted, because the smart farm sector mainly implements convergent and applied technology, R&D actors' features are not prominent. Therefore, we conduct a comparative analysis using the overall weight range, rather than differentiating between individual weights by type of actor.

Under the unconstrained DEA model, multi-sensing AI control (T13) and hybrid renewable energy (T16) have the highest efficiency (1.000). In contrast, in the weighted DEA-AR model, only hybrid renewable energy (T16) has a high efficiency of one. Furthermore, after setting the constraints, most DMUs are less efficient than they were in the unconstrained DEA. Given efficiency scores between the DEA and DEA-AR model in Fig 3, the following 8 DMUs get worse by applying the constraints: smart machinery (T7), smart product sorting (T10), long-term storage packaging technology (T11), Multi-sensing AI control (T13), artificial synthesis (T14), Nano-bio sensing (T19), smart food (T26). It implies that a DMU in the unrestricted virtual price model identified as efficient can be turned out to be inefficient in the reality that analysts can assign subjective value judgements on the relative proportions of virtual prices. For example, Multi-sensing AI control (T13) is the most efficient technology in basic DEA model, but its efficiency gets worse after reflecting opinions in the related field. If policymakers decide R&D resource allocation only by efficiency, Multi-sensing AI control (T13) and hybrid renewable energy (T16) would be selected. However, Multi-sensing AI control (T13) is actually less efficient in reality and it is possible that other efficient technologies have been excluded. Therefore, it implies that reflecting field opinions is crucial when making resource allocation decision for policymakers.

Comparing the rankings of the DMUs (Table 6), the following 10 rank higher after applying the weights: environmental condition monitoring (T1), watering system measurement (T3), smart irrigation system (T4), cooling/heating AC system control (T5), biometric information measurement (T6), pest and disease diagnosis (T8), smart product sorting (T10), customized functional ingredient control (T18), smart quality control and distribution platform (T23), and advanced production (T25). Multi-sensing AI control (T13) ranks first when is no restriction; however, drops to seventh when using the DEA-AR model. The following DMUs do not change rank: pest and disease management (T9), vertical farm building design (T12), artificial synthesis (T14), integrated environmental information processing platform (T20), crop growth big data platform (T21), and productivity optimizing management platform (T22).

As mentioned earlier, in this study, there is a case where there is no consistency between the upper and lower bounds for the ratio of relative prices added as constraints of AR, and the consistency is corrected (the lower bound is adjusted from 0.5 to 1) to perform a sensitivity analysis. As a result of carrying out the sensitivity analysis, almost the same outcomes were obtained as in the case without correction. This result was interpreted to show that the level of lack of consistency does not show much difference compared to the size of the relative upper bound.

**Comparative analysis in terms of government investment.** Table 6 shows the technology groups selected as investment priorities for two years (2019–2020), as well as the DEA and DEA-AR efficiency rankings estimated in this study. According to Table 6, of the 14 technologies selected as investment priorities, five ranked higher in the DEA-AR model: cooling/heating AC system control (T5), from 16 to 14; smart product sorting (T10), from 3 to 2; customized functional ingredient control (T18), from 17 to 13; smart quality control and distribution platform (T23), from 19 to 18; and advanced production (T25), from 15 to 12. The investment priorities of AI farmbot (T15) and hybrid renewable energy (T16) do not change. AI farmbot (T15) has a lower efficiency ranking, owing to the weight constraints, but hybrid renewable energy (T16) remains unchanged. These results are assumed to reflect reality when

**Table 6. Results of investment priorities from 2019 to 2020.**

| DMU | Technology Group | DEA Rank | DEA-AR Rank | Investment Priority |
|-----|------------------|----------|-------------|---------------------|
| T1 | Environmental Condition Monitoring | 11 | 9 | |
| T2 | Smart Complex Environmental Controls | 18 | 20 | |
| T3 | Watering System Measurement | 8 | 4 | |
| T4 | Smart Irrigation System | 7 | 3 | |
| T5 | Cooling/Heating AC System Control | 16 | 14 | O |
| T6 | Biometric Information Measurement | 9 | 5 | |
| T7 | Smart Machinery | 5 | 11 | O |
| T8 | Pests and Disease Diagnosis | 20 | 15 | |
| T9 | Pests and Diseases Management | 10 | 10 | |
| T10 | Smart Product Sorting | 3 | 2 | O |
| T11 | Long-term Storage Packaging Technology | 4 | 8 | |
| T12 | Vertical Farm Building Design | 21 | 21 | |
| T13 | Multi-sensing AI Control | 1 | 7 | O |
| T14 | Artificial Synthesis | 6 | 6 | |
| T15 | AI Farmbot | 13 | 17 | ◎ |
| T16 | Hybrid Renewable Energy | 1 | 1 | O |
| T17 | Organic Water Reclamation | N/A | N/A | O |
| T18 | Customized Functional Ingredient Control | 17 | 13 | O |
| T19 | Nano-bio-sensing | 14 | 19 | |
| T20 | Integrated Environmental Information Processing Platform | 23 | 23 | O |
| T21 | Crop Growth Big Data Platform | 24 | 24 | |
| T22 | Productivity Optimizing Management Platform | 22 | 22 | O |
| T23 | Smart Quality Control and Distribution Platform | 19 | 18 | O |
| T24 | Printed Farm | N/A | N/A | O |
| T25 | Advanced Production | 15 | 12 | O |
| T26 | Smart Food | 12 | 16 | O |

Note:

◎ indicates overlapping investment areas for two years.

aligning the weight to the AHP. Hybrid renewable energy (T16), integrated environmental information processing platform (T20), and productivity optimizing management platform (T22) remain unchanged after the weight constraint is applied. Furthermore, organic water reclamation (T17) and printed farm (T24), which have no cumulative investment (zero input value), were included in the investment area. The fact that hybrid renewable energy (T16) was selected as the most efficient technology group in the DEA and DEA-AR models indicates that working-level opinions are considered when making investment decisions.

**Sensitivity analysis results.** We performed sensitivity analysis as well to show how stable the analysis are while gradually changing the inputs. As suggested in the previous study, firstly, DMUs were classified by efficiency (efficient or inefficient) and then the input value was changed form 5% to 20%. We introduce 5% to 20% increases in each of input for T16. Simultaneously, we decreases these inputs by 5% to 20% for the other (inefficient) DMUs. Although T16, an efficient DMU, changed from 5% to 20%, it was always an efficient DMU while the inefficient DMUs did not change. Figs 4–7 show the results of sensitivity and stability analysis in DEA-AR model by professional group. As stated in previous study, these results

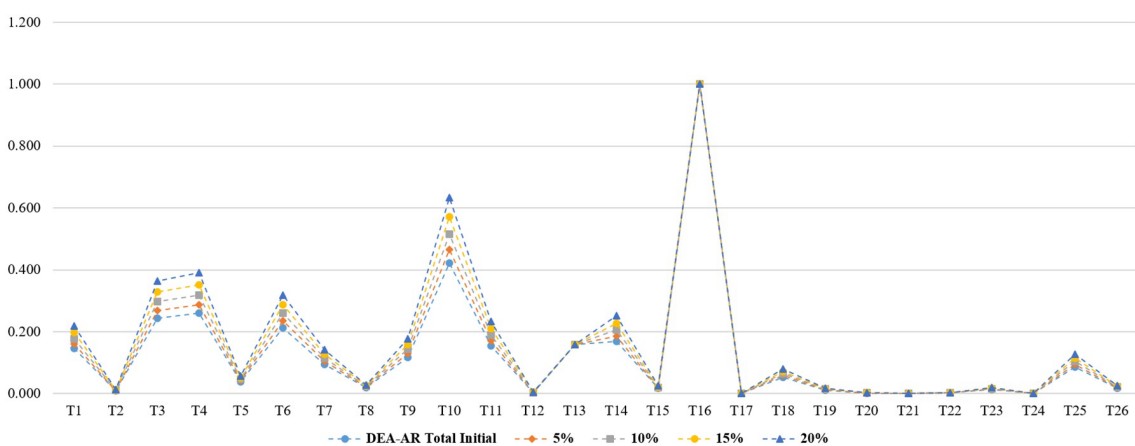

**Fig 4. Sensitivity and stability analysis in DEA-AR model (total).**

significantly increase the robustness of the proposed DEA model. In other words, it was confirmed whether the results of this research were empirically robust through sensitivity analysis.

## Discussion

The 26 smart farming technology groups use convergence technologies, in which application technology is employed rather than basic technology. These characteristics are also common to the weighting results of the R&D actors. The basic research-oriented academia and research institutes that serve as a bridge between basic research and applied development emphasize high weights on patents, the employment inducement effect, the value-added inducement effect, and the forward and backward industry link effects. That is, even for academia and research institutes, where basic technology is considered theoretically important, the performance of technology application and development is as significant as it is in industry. This implies that reflecting the opinions of such actors on R&D performance indicators is crucial to selecting the appropriate investment decision model. In the DEA-AR model, which applies the

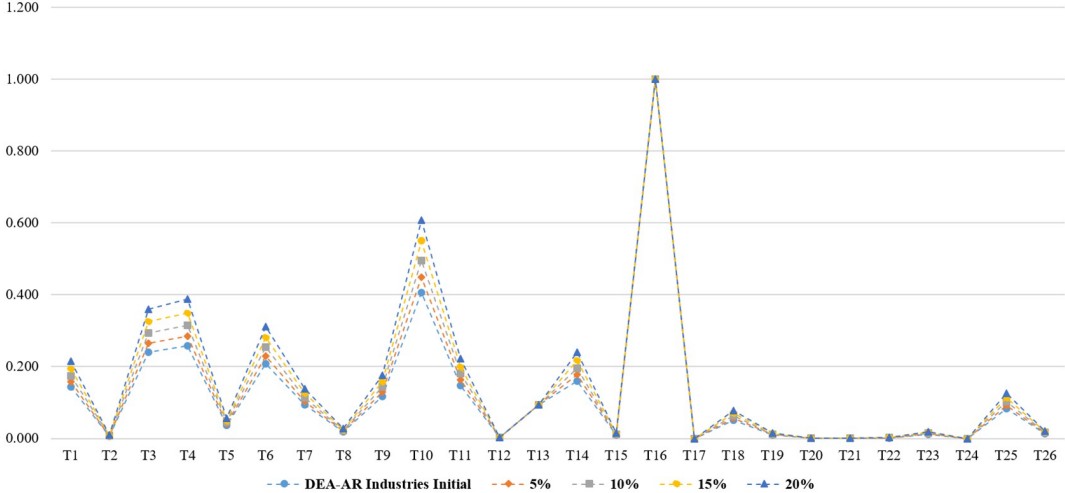

**Fig 5. Sensitivity and stability analysis in DEA-AR model (industries).**

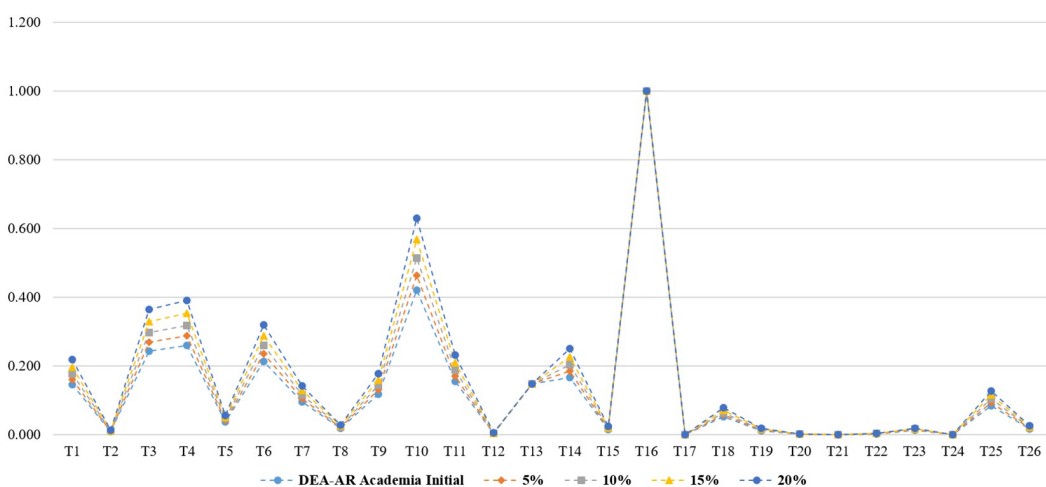

**Fig 6. Sensitivity and stability analysis in DEA-AR model (academia).**

opinions of the R&D actors, the opinions for the individual R&D actors are similar to the overall viewpoint, indicating that the subjective judgements of each R&D actor do not have a significant effect on investment decision-making in the smart farm sector.

The government selected investment priorities for two consecutive years (from 2019 to 2020), mainly consisting of IoT platform technologies and AI control and sensing technologies. Thus, when the government decides to allocate R&D investments, they recognize the importance of core technologies in the field. Furthermore, a comparison of the investment priorities for the two years from the DEA-AR model (reflecting the views of experts) and the DEA model (virtual price is calculated automatically) shows that technologies are selected that have a relative higher ranking or that are at the same level. Among the 26 technology groups, hybrid renewable energy (T16) was the most efficient DMU in both models. Therefore, hybrid renewable energy (T16) is a decision that meets the investment needs of reality. However, other technology groups are not considered investment priorities, owing to a low theoretical efficiency; however, their relative efficiency increases after reflecting the opinions of experts, that is T1 (environmental condition monitoring), T3 (watering system measurement), T4

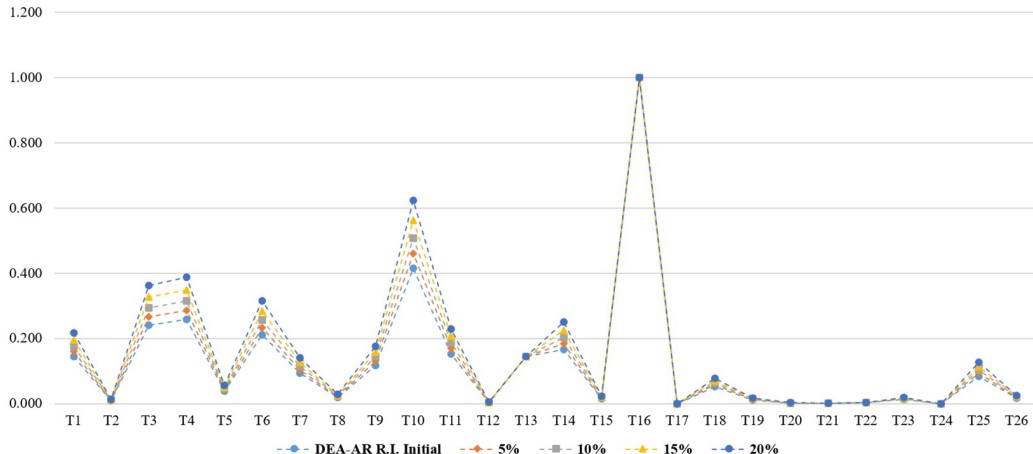

**Fig 7. Sensitivity and stability analysis in DEA-AR model (research institutes).**

(smart irrigation system), T6 (Biometric information measurement), and T8 (pest and disease diagnosis). In reality, it is necessary to invest in these groups, but they are excluded from top-down national R&D investment decisions, which distorts the investment allocations. In addition, additional detailed analyses are needed to determine whether the technology groups selected for investment are relatively efficient compared with those that are not selected. Moreover, we need to reconsider the policy of which technology groups are selected for investment from among relatively low- and high-rank technology groups. This will improve the investment efficiency when making investment allocation decisions from policymakers' points of view. Therefore, we need a methodology that quantitatively reflects experts' views in the field related to performance measurement and decision-making for national R&D investment efficiency. Previous studies regarding DEA mainly compare the DEA results among DMUs. However, our study reflect the characteristics of smart farm industry based on R&D stakeholders' opinions. Therefore, the proposed DEA-AR model can be used as a methodology to bridge the gap between the theory and the reality, and can compensate for quantitative distortions caused by top-down national R&D investment decisions.

The difference from previous studies is that we made the most of the advantages of the DEA model by preemptively reflecting the importance of performance indicators for each R&D actor in the process of allocating the national R&D budget. This study also contributes to the fact that it objectifies national R&D investment allocation decisions and secures reliability for investment alternative preferences through additional sensitivity analysis.

## Conclusion

This study suggests a methodology for modeling R&D investment efficiency that incorporates the opinions of R&D actors and can be used for government decision-making related to R&D planning and investments. A conventional DEA analysis is meaningful in recommending the most efficient and optimal DMU, automatically and theoretically. However, in addition to providing theoretical results, the DEA-AR model presented here uses actors' opinions to constrain the range of the weights, thus providing reasonable and objective conclusions. The AHP, based on the principles of human behavior, is a problem-solving decision-making method that combines subjective judgment with systemic approaches. That is, we calculate the weights based on experts' opinions by considering important key performance R&D indicators that adequately reflect their opinions. Moreover, the DEA ensures the objectivity of the analysis of the top-down national R&D investment decision-making process, which mainly reflects the subjective opinions of policymakers. In particular, by applying a DEA-AR model instead of a general DEA model, we solve the latter's problem of low discriminating power by restricting the weights. In other words, it is possible to objectively judge whether national R&D investment allocations and planning are consistent with reality using rational weight values from experts in each R&D sector in an AHP analysis. Therefore, the DEA-AR model provides flexible and unbiased national R&D investment decisions that reflect the actual situation and intention of internal and external R&D planning. The model can also be used to implement policies for strategic decisions, as well as to improve future investment efficiency.

We examined the importance of key performance indicators for each R&D actor, considering the case of Korean smart farms, which represent one of 10 key areas identified by the Korean government. The study also proposes a national R&D investment decision-making model that reflects the industrial smart farm characteristics of each R&D actor (i.e., academia, industry, and research institutes) which is a new qualitative and quantitative investment alternative analysis model. The range of weights (or preferences) for performance indicators was analyzed using AHP by collecting R&D actor's opinions. In addition, efficiency analysis was

performed by reflecting the range of weights for performance indicators as additional constraints. This can be said to have suggested a method to objectively quantify and reflect expert opinions in national R&D investment decision making. Finally, in order to analyze the category in which the efficiency score for each investment alternative does not change, sensitivity analysis was performed to secure the reliability of the proposed model. The model we proposed emphasizes practicality so that it can be used in the actual decision making process. AHP was applied as a method to systematically supplement the problem of discriminating power that may be pointed out as a limitation of the conventional DEA model while making the most of the objectivity of the DEA model. Our approach can be extended to various policy decision making issues that should properly reflect the decision maker's willingness to evaluate while maintaining the objectivity of the decision making process in a special situation such as a top down manner. In addition, in practical terms, this model is basically based on the linear programming, so it is considered relatively easy to apply to other cases.

This study is conducted for the short term (i.e., two years), owing to a lack of available data. However, national R&D investment efficiency and its performance need to be analyzed over a longer period, and the smart farm technology groups (DMUs) used in this analysis may change over time. Therefore, additional variables are required. We can also apply a model other than the AHP to constrain the weight ranges. Furthermore, there is room to extend this model to other major national R&D investment fields. These topics are left to future research.

## Supporting information

**S1 Table. Comparison of efficiency scores and ranks between the DEA and DEA-AR models.**
(PDF)

**S2 Table. Results of the DEA and DEA-AR model analyses by R&D stakeholders.**
(PDF)

## Acknowledgments

The authors would like to thank the Editor-in-Chief and anonymous reviewers for their helpful comments and suggestions.

## Author Contributions

**Conceptualization:** Wan Park, Sang-Gook Kim.

**Data curation:** Wan Park.

**Formal analysis:** Wan Park, Sang-Gook Kim.

**Investigation:** Wan Park.

**Methodology:** Wan Park, Sang-Gook Kim.

**Software:** Wan Park, Sang-Gook Kim.

**Supervision:** Sang-Gook Kim.

**Validation:** Wan Park, Sang-Gook Kim.

**Writing – original draft:** Wan Park.

**Writing – review & editing:** Wan Park, Sang-Gook Kim.

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
