## [Decision Letter · Decision Letter 0]

3 Dec 2021

PONE-D-21-35012Integrating quantitative and qualitative methodologies to build a national R&D plan using Data Envelopment Analysis based on R&D stakeholders’ perspectivesPLOS ONE

Dear Dr. Kim,

Thank you for submitting your manuscript to PLOS ONE. After careful consideration, we feel that it has merit but does not fully meet PLOS ONE’s publication criteria as it currently stands. Therefore, we invite you to submit a revised version of the manuscript that addresses the points raised during the review process.

We look forward to receiving your revised manuscript.

Kind regards,

Yang Li

Academic Editor

PLOS ONE

Journal Requirements:

" ext-link-type="uri" xlink:type="simple">https://journals.plos.org/plosone/s/file?id=ba62/PLOSOne_formatting_sample_title_authors_affiliations.pdf"

Reviewers' comments:

Reviewer's Responses to Questions

**Comments to the Author**

1. Is the manuscript technically sound, and do the data support the conclusions?

Reviewer #1: Yes

Reviewer #2: Yes

2. Has the statistical analysis been performed appropriately and rigorously? 

Reviewer #1: Yes

Reviewer #2: Yes

3. Have the authors made all data underlying the findings in their manuscript fully available?

Reviewer #1: Yes

Reviewer #2: Yes

4. Is the manuscript presented in an intelligible fashion and written in standard English?

Reviewer #1: Yes

Reviewer #2: Yes

5. Review Comments to the Author

Reviewer #1: Generally speaking, this paper is well written and technically sound. Here, there are some concerns of this reviewer:

1 The computational cost of the proposed approach should be discussed.

2 More jobs should be done to verify the contributions of this work.

3 Authors have not presented the limitations of this work. How this work can be extended in practical applications? Please elaborate on it.

4 when using the presented method, the motivation of the combination of analytic hierarchy process (AHP) and DEA model should be better justified.

5 All references, especially article titles, should be in a uniform format. Please check and modified the references.

Reviewer #2: This paper presents the integrating quantitative and qualitative methodologies to build a national RD plan by using data envelopment analysis. The test results are also provided. In general, the topic is interesting. I think that the quality of this paper can be improved if the authors address the following aspects:

1 The main contributions and novelty of this paper should be further summarized. This reviewer suggests the authors exactly mention what is new compared with existing approaches and why the proposed approach is needed to be used instead of the existing methods. This reviewer suggests the authors use bullets (3 bullets is standard) and in each bullet explain one contribution clearly.

2 How scalable is the presented approach?

3 The proposed method might be sensitive to the values of its main controlling parameters. How did you tune the parameters? Please elaborate on it.

4 Authors need to add more results to thoroughly support the main findings.

5 Analytic hierarchy process (AHP) is an important item in the proposed approach. Some studies regarding applications of AHP in engineering like [doi.org/10.1155/2016/3965608] can be cited to improve the literature survey.

6. PLOS authors have the option to publish the peer review history of their article (what does this mean?). If published, this will include your full peer review and any attached files.

Reviewer #1: No

Reviewer #2: No

---

## [Author Response · Author response to Decision Letter 0]

17 Jan 2022

Reviewer #1: Generally speaking, this paper is well written and technically sound. Here, there are some concerns of this reviewer:

1. The computational cost of the proposed approach should be discussed. 

Response: The computational cost in DEA is highly related to the number of input and output variables. In the investment efficiency analysis model for national RD proposed in this study, the number of input and output variables is generally not large. Similarly, in the smart farms case, the input variable is RD budget and the output is composed of performance indicators that are technical and economic effect. Reflecting the reviewer’s suggestion, we performed the sensitivity analysis to secure the reliability of the proposed model. The reason for this is to confirm whether the complementary slackness theory of the proposed DEA-AR model is satisfied within a specific change range of the input. If the conversion to the dual problem is possible within the range of the amount of change in the input that satisfies the complementary margin condition, there is a possibility that the computational cost may be lowered. Data transformation within the range of maintaining an optimal solution in DEA (i.e., radii of stability) does not change the efficiency of the DMU. When the complementary slackness condition is satisfied, the envelop model and the virtual model have a dual relationship, and the solution is the same. Therefore, if the complementary slackness condition is satisfied, there is room for the computational cost to be reduced by the number of variables of the objective function. In this study, sensitivity analysis was additionally performed to examine whether the complementary slackness condition were met. According to previous study [1], the sensitivity analysis of DEA requires a new algorithm because the method used in the existing linear programming is not suitable to be used as a constraint of the linear programming model used in DEA. Sensitivity analysis was performed to show to what extent the analysis results were stable while gradually changing the inputs by referring to previous study. As suggested in the previous literature, we confirmed the stability of the analysis results by changing inputs from 5% to 20% for efficient and inefficient DMUs (please refer to the reviewer’s third question for more detailed results). Therefore, the proposed model in this study also satisfies the complementary slackness condition and enables dual problem solving between constraints and objective variables, which reducing computational cost. However, the given constraints and the number of variables in the original problem are the same and additionally constrained by the assurance region is calculated as primal not dual because it limits only the feasible range of region.

Refer to page number of revised manuscript with track changes : pp.19-20 

2. More jobs should be done to verify the contributions of this work. 

Response: The contribution of this study can be summarized into three categories. First, a scale was developed to measure efficiency in calculating various performances compared to RD expenditures in the budget allocation process of national RD. In particular, a new qualitative and quantitative analysis was presented to measure the efficiency of input to performance within the range of importance for output variables intended by industry, academia, and research institutions. As a result, this study is meaningful in that it makes the most advantages of the DEA model and objectifies the national RD investment decision making in a top down manner. Second, various attempts have been made to specify assurance region in DEA-AR model, which limits the range of values that a weight can have by adding constraints to DEA. The difference from previous studies is that the importance (or preference) of investment priority performance indicators by innovation entities (i.e. industry, academia, and research institutes) were preemptively evaluated through AHP, and the evaluation results were constructed and reflected in DEA model. Finally, this study contributes to the fact that it is necessary to analyze the change in the appropriate input that the optimal investment priority will not change, emphasizing the necessity of performing sensitivity analysis to secure the reliability of the proposed DEA-AR model. 

Fig 1. The revised conceptual analytical model

Refer to page number of revised manuscript with track changes : p.8 

3. Authors have not presented the limitations of this work. How this work can be extended in practical applications? Please elaborate on it. 

Response: The limitations of this study were briefly mentioned in the conclusion to two (p.27). First, national RD investment efficiency and its performance need to be analyzed over a longer period, and the smart farm technology groups (DMUs) used in this analysis may change over time. Therefore, additional variables are required. Secondly, it is necessary to apply a model other than the AHP to constrain the weight ranges. 

(p. 32) This study is conducted for the short term (i.e., two years), owing to a lack of available data. However, national RD investment efficiency and its performance need to be analyzed over a longer period, and the smart farm technology groups (DMUs) used in this analysis may change over time. Therefore, additional variables are required. We can also apply a model other than the AHP to constrain the weight ranges. Furthermore, there is room to extend this model to other major national RD investment fields. These topics are left to future research.

In particular, this study does not focus on methodology itself, but combines DEA and AHP which have been empirically conducted based on previous studies, so that decisions can be made based on quantitative data by reflecting the opinions of experts in the field. Therefore, if the proposed approach is applied and extended to other major national RD fields, it is expected that more valid investment allocation decisions will be made reflecting the characteristics of each field. 

Following the reviewer’s opinion, we additionally performed the sensitivity analysis to secure the stability of proposed DEA-AR model in this study. Cooper et.al. [1] studied some developments of sensitivity and stability analysis in DEA. They noted that the standard approaches for carrying out sensitivity analysis in linear programming could not apply to DEA. So they mentioned that a new algorithm is needed for the DEA sensitivity analysis. Therefore, we performed sensitivity analysis as well to show how stable the analysis are while gradually changing the inputs. As suggested in the previous study, firstly, DMUs were classified by efficiency (efficient or inefficient) and then the input value was changed form 5% to 20%. We introduce 5% to 20% increases in each of input for T16. Simultaneously, we decreases these inputs by 5% to 20% for the other (inefficient) DMUs. Although T16, an efficient DMU, changed from 5% to 20%, it was always an efficient DMU while the inefficient DMUs did not change. Figure 4 to 78 show the results of sensitivity and stability analysis in DEA-AR model by professional group. 

Fig 4. Sensitivity and stability analysis in DEA-AR model (total)

Fig 5. Sensitivity and stability analysis in DEA-AR model (industries)

Fig 6. Sensitivity and stability analysis in DEA-AR model (academia)

Fig 7. Sensitivity and stability analysis in DEA-AR model (research institutes)

As stated in previous study, these results significantly increase the robustness of the proposed DEA model. In other words, it was confirmed whether the results of this research were empirically robust through sensitivity analysis. Therefore, if sensitivity analysis is extended in practical application in the future, it will be possible to identify conditions (or variables) that may be sensitive to make investment allocation decisions. 

 Refer to page number of revised manuscript with track changes : pp.19-20, p.27-29 

4. When using the presented method, the motivation of the combination of analytic hierarchy process (AHP) and DEA model should be better justified. 

Response: One of the biggest advantages of DEA is that the analyst does not provide prior information regarding the value or weight of input and output factor respectively, but finds a theoretical optimal solution in the model. Since it has a rule to set the virtual price where the efficiency of observation becomes the highest, complaints whether the weight setting is well established cannot occur. All DEA models, such as CCR and BCC, are eligible to apply the most favorable virtual price without limitation from the observations. However, if the researcher knows anything in advance about the input or output factor, it may impose constraints so that the calculated virtual price conforms to the subjective value judgment. For instance, when allocating and investing a limited national RD budget as in this study, it is a rational decision to allocate the budget to a technology group with high performance (i.e. output) compared to the investment (i.e. input). If the policymaker knows in advance the value of input or output factors, it may be realistic to constrain the calculated virtual price beforehand so that it is consistent with this subjective value judgment. In addition, if experts’ judgment value is reflected in advance to make rational decisions, it can enhance the reality of the results. For this problem, various types of virtual price constraints have been proposed in previous literatures [2]. Thus, our study aims to secure the rationality of policy decision making by quantitatively reflecting the qualitative opinions of experts. In particular, this study adds constraints (i.e. assurance region) by strategically reflecting the expert’s intentions when planning national RD to the model so that it may solve the problem of discriminating power where the DEA model itself contains. Among the various methods of calculating weights for constraints, we applied AHP as it is already being used to analyze complex decision making problems in many fields such as policy alternative analysis and selection, strategic decision making problems, RD planning and feasibility studies, and RD performance analysis. AHP is one of the decision making methodology that provides a framework for analysis to solve a number of conflicting decision making issues in a systematic manner. In this study as well, we were able to reflect the reasonable weight of experts by RD subject through comparison between factors by layering the decision making structure of experts. 

Refer to page number of revised manuscript with track changes : p.5 

5. All references, especially article titles, should be in a uniform format. Please check and modified the references.

Response: We used the ‘Mendeley’ program to list up the references, style Plos One. As your recommendation, we double checked thoroughly and modified article titles in a uniform format manually. We thought the program did not change title automatically. We appreciated your comments. 

Refer to page number of revised manuscript with track changes : pp.33-36

References

1. Cooper WW, Li S, Seiford LM, Tone K, Thrall RM, Zhu J. Sensitivity and stability analysis in DEA: some recent developments. J Product Anal. 2001;15: 217–246. doi:10.1023/A:1011128409257

2. R.G.Dyson, E.Thanassoulis. Reducing weight flexibility in data envelopment analysis. J Oper Res Soc. 1988;39: 563–576. 

 

Reviewer #2: This paper presents the integrating quantitative and qualitative methodologies to build a national RD plan by using data envelopment analysis. The test results are also provided. In general, the topic is interesting. I think that the quality of this paper can be improved if the authors address the following aspects: 

1. The main contributions and novelty of this paper should be further summarized. This reviewer suggests the authors exactly mention what is new compared with existing approaches and why the proposed approach is needed to be used instead of the existing methods. This reviewer suggests the authors use bullets (3 bullets is standard) and in each bullet explain one contribution clearly. 

Response: The contribution of this study can be summarized as follows. 

A scale was developed to measure efficiency in calculating various performances compared to RD expenditures in the budget allocation process of national RD. In particular, a new qualitative and quantitative analysis was presented to measure the efficiency of input to performance within the range of importance for output variables intended by industries, academia, and research institutions. As a result, this study is meaningful in that it makes the most advantages of the DEA model and objectifies the national RD investment decision making in a top down manner. As a result, this study is meaningful in that it makes the most advantages of the DEA model and objectifies the national RD investment decision making in a top down manner. The characteristic of the DEA model is that it enables to objectively determine the efficiency or inefficiency of various organizations or business with minimal information. Especially, there are many evaluation criteria when evaluating RD projects to determine the allocation of RD investment. It is a common method to preemptively assign weights to the importance of each evaluation criterion as a way to integrate these criteria. However, a priori weighting on the importance of evaluation is essentially a matter of subjective judgement. In this regard, there is a possibility that the lack of objectivity may undermine the objectivity of the entire decision making process. DEA does not necessarily require a priori weighting among evaluation criteria, and even when the weights are added to the model, only an interval value rather than a single value is sufficient. Therefore, DEA can be seen as one of the models that can give objectivity to the national RD investment allocation decision making process. 

Various attempts have been made to specify assurance region in DEA-AR model, which limits the range of values that a weight can have by adding constraints to DEA. Considering studies that incorporate AHP into DEA, AHP is applied to factors that cannot be quantified among DEA inputs and outputs [1], or selects DMU rankings [2], or derives a comparative matrix using a DEA to design a method for ranking alternatives using an AHP [3]. The difference from previous studies is that the importance (or preference) of investment priority performance indicators by innovation entities (i.e. industry, academia, research institutes) were preemptively evaluated through AHP, and the evaluation results were constructed and reflected in DEA model. In addition, following the reviewer’s comment, we performed sensitivity analysis additionally to confirm the stability of the model, which has not been further conducted in the previous literatures. As a result, we confirmed the stability of the proposed model. 

It is necessary to conduct the sensitivity analysis to secure the reliability (or stability) of the proposed DEA-AR model in which the qualitative and quantitative analysis are mixed. At the same time, this study contributes to the fact that it is necessary to analyze the change in the appropriate input that the optimal investment priority will not change (please refer to the reviewer’s third question for more detailed sensitivity analysis results).

Fig 1. The revised conceptual analytical model

Refer to page number of revised manuscript with track changes : p.5

2. How scalable is the presented approach? 

Response: The presented approach is a model that emphasizes practicality so that it can be used in the actual decision making process. The characteristics of the proposed model is that it utilizes the feature of objectivity of DEA model as much as possible while incorporating AHP as a method to systematically supplement the problem of discriminating power which can be pointed out as a limitation of DEA. Besides, we proposed a method to objectively quantify and reflect expert opinions inside the model. The approach we presented can be extended and applied to various policy decision making cases in which reflect the decision makers’ willingness while maintaining the objectivity of the decision making process in a special decision making situation in the future. Also, from a practical point of view, this model is basically based on the linear programming, so it is relatively easy to apply to other studies. Additionally supplemented sensitivity analysis is necessary to measure the stability of the output according to the change of the input. In addition, it can play an important role in identifying investment alternatives as an important means in RD investment decision making. 

Refer to page number of revised manuscript with track changes : p.8, p.13, pp.19-20, pp.27-29

3. The proposed method might be sensitive to the values of its main controlling parameters. How did you tune the parameters? Please elaborate on it. 

Response: DEA measures the efficiency of output factors compared to input factors nonparametrically. After deriving empirical efficiency frontier using the data between the empirical inputs and outputs, the inefficiency is measured by the distance between the DMUs from the efficient frontier. The efficiency level of observations (DMUs) is calculated. In order to improve efficiency, information on which producers should be targeted as a benchmark and how to adjust inputs and outputs is obtained. All these processes are not ‘estimated’ probabilistically by statistical assumptions or analyses, but are definitively ‘calculated’ as a solution to the linear programming. The conventional DEA has the advantages as a non-statistical attribute in that the analysis procedure is relatively simple and the researcher’s arbitrary judgement is excluded because it does not require statistical assumptions. In other words, since it is a methodology that considers the relative ratio of input to output, sensitivity to the controlling parameter is not inherently considered. However, in this study, a sensitivity analysis was additionally performed to confirm the stability of our results to reflect the reviewer’s opinion. According to previous [4], the standard approaches for carrying out sensitivity analysis in linear programming could not apply to DEA. So it is mentioned that a new algorithm is needed for the DEA sensitivity analysis. Therefore, we performed sensitivity analysis as well to show how stable the analysis are while gradually changing the inputs. As suggested in the previous study, firstly, DMUs were classified by efficiency (efficient or inefficient) and then the input value was changed form 5% to 20%. We introduce 5% to 20% increases in each of input for T16. Simultaneously, we decreases these inputs by 5% to 20% for the other (inefficient) DMUs. Although T16, an efficient DMU, changed from 5% to 20%, it was always an efficient DMU while the inefficient DMUs did not change. Figure 5 to 8 show the results of sensitivity and stability analysis in DEA-AR model by professional group. As stated in previous study, these results significantly increase the robustness of the proposed DEA model. In other words, it was confirmed whether the results of this research were empirically robust through sensitivity analysis.

Fig 5. Sensitivity and stability analysis in DEA-AR model (total)

Fig 6. Sensitivity and stability analysis in DEA-AR model (industries)

Fig 7. Sensitivity and stability analysis in DEA-AR model (academia)

Fig 8. Sensitivity and stability analysis in DEA-AR model (research institutes)

Refer to page number of revised manuscript with track changes : p.8, p.13, pp.19-20, pp.27-29

4. Authors need to add more results to thoroughly support the main findings. 

Response: This study newly proposed a qualitative and quantitative investment alternative analysis model which is necessary to make a decision in national RD investment. At this time, the range of weights (or preferences) for performance indicators is analyzed using AHP by collecting expert opinions for each innovation entities (i.e. industry, academia, and research institutions). After that, efficiency analysis is performed by reflecting the range of weights for performance indicators in an additional constraint in the DEA model. Finally, in order to analyze the category in which the efficiency score for each investment alternative does not change, sensitivity analysis was performed to secure the reliability of the newly proposed model (please refer to the reviewer’s third question for more detailed sensitivity analysis results).

Refer to page number of revised manuscript with track changes : p.5, p.8, p.13, pp.19-20, pp.27-29

5. Analytic hierarchy process (AHP) is an important item in the proposed approach. Some studies regarding applications of AHP in engineering like [doi.org/10.1155/2016/3965608] can be cited to improve the literature survey.

Response: There is a room for other AHP methodologies to be applied as long as it solidifies our research. As you recommended we reviewed the literature suggested but there is any relevant to our study and we consider this inclusion would not improve the manuscript. We would like to show you our appreciation in your recommendation fully and in the near future if we do another studies regarding AHP applications we will review the literatures including your proposed thoroughly. 

References

1. Sueyoshi T, Shang J, Chiang WC. A decision support framework for internal audit prioritization in a rental car company: A combined use between DEA and AHP. Eur J Oper Res. 2009;199: 219–231. doi:10.1016/j.ejor.2008.11.010

2. Sinuany-Stern Z, Mehrez A, Hadad Y. An AHP/DEA methodology for ranking decision making units. Int Trans Oper Res. 2000;7: 109–124. doi:10.1111/j.1475-3995.2000.tb00189.x

3. Ramanathan R. Data envelopment analysis for weight derivation and aggregation in the analytic hierarchy process. Comput Oper Res. 2006;33: 1289–1307. doi:10.1016/j.cor.2004.09.020

4. Cooper WW, Li S, Seiford LM, Tone K, Thrall RM, Zhu J. Sensitivity and stability analysis in DEA: some recent developments. J Product Anal. 2001;15: 217–246. doi:10.1023/A:1011128409257

---

## [Decision Letter · Decision Letter 1]

23 Feb 2022

Integrating quantitative and qualitative methodologies to build a national RD plan using Data Envelopment Analysis based on RD stakeholders’ perspectives

PONE-D-21-35012R1

Dear Dr. Kim,

We’re pleased to inform you that your manuscript has been judged scientifically suitable for publication and will be formally accepted for publication once it meets all outstanding technical requirements.

Kind regards,

Dragan Pamucar

Academic Editor

PLOS ONE

Additional Editor Comments (optional):

Reviewers' comments:

Reviewer's Responses to Questions

**Comments to the Author**

1. If the authors have adequately addressed your comments raised in a previous round of review and you feel that this manuscript is now acceptable for publication, you may indicate that here to bypass the “Comments to the Author” section, enter your conflict of interest statement in the “Confidential to Editor” section, and submit your "Accept" recommendation.

Reviewer #1: All comments have been addressed

Reviewer #3: All comments have been addressed

2. Is the manuscript technically sound, and do the data support the conclusions?

Reviewer #1: Yes

Reviewer #3: Yes

3. Has the statistical analysis been performed appropriately and rigorously? 

Reviewer #1: Yes

Reviewer #3: Yes

4. Have the authors made all data underlying the findings in their manuscript fully available?

Reviewer #1: Yes

Reviewer #3: Yes

5. Is the manuscript presented in an intelligible fashion and written in standard English?

Reviewer #1: Yes

Reviewer #3: Yes

6. Review Comments to the Author

Reviewer #1: Authors have fully considered my comments on the previous version of the paper. I'm satisfied with the modifications made in the revised version. Therefore, I think this paper deserves to be published in its current form. Congratulations.

Reviewer #3: The authors have addressed the point of my concern. I am happy with their corrections. Hence, I would like to recommend this manuscript to be published.

7. PLOS authors have the option to publish the peer review history of their article (what does this mean?). If published, this will include your full peer review and any attached files.

Reviewer #1: No

Reviewer #3: No

---

## [Editor Report · Acceptance letter]

28 Feb 2022

PONE-D-21-35012R1 

Integrating quantitative and qualitative methodologies to build a national RD plan using Data Envelopment Analysis based on RD stakeholders’ perspectives 

Dear Dr. Kim:

I'm pleased to inform you that your manuscript has been deemed suitable for publication in PLOS ONE. Congratulations! Your manuscript is now with our production department. 

Kind regards, 

on behalf of

Dr. Dragan Pamucar 

Academic Editor

PLOS ONE